# Soliton confinement in a quantum circuit

Ananda Roy [1] ✉ & Sergei L. Lukyanov[1] ✉

Confinement of topological excitations into particle-like states - typically associated with theories of elementary particles - are known to occur in condensed matter systems, arising as domain-wall confinement in quantum spin chains. However, investigation of confinement in the condensed matter setting has rarely ventured beyond lattice spin systems. Here we analyze the confinement of sine-Gordon solitons into mesonic bound states in a perturbed quantum sine-Gordon model. The latter describes the scaling limit of a one-dimensional, quantum electronic circuit (QEC) array, constructed using experimentally-demonstrated QEC elements. The scaling limit is reached faster for the QEC array compared to spin chains, allowing investigation of the strong-coupling regime of this model. We compute the string tension of confinement of sine-Gordon solitons and the changes in the low-lying energy spectrum. These results, obtained using the density matrix renormalization group method, could be verified in a quench experiment using state-of-the-art QEC technologies.

Confinement and asymptotic freedom are paradigmatic examples of non-perturbative effects in strongly interacting quantum field theories (QFTs)[1]. While typically associated with theories of elementary particles[2,3], confinement of excitations into particle-like states occurs in a wide range condensed matter systems. In the latter setting, the "hadrons" are formed due to confinement of domain walls in quantum spin chains[4]. They have been detected using neutron scattering experiments in a coupled spin-1/2 chains[5] and in a one-dimensional Ising ferromagnet[6]. Furthermore, signatures of confinement have been observed in numerical investigations of quenches in quantum Ising spin chains[7,8] as well as in noisy quantum simulators[9,10].

Despite its ubiquitousness, in the condensed matter setting, quantitative investigation of confinement has rarely ventured beyond lattice spin systems. In this work, we show that confinement of topological excitations can arise in a one-dimensional, superconducting, quantum electronic circuit (QEC) array. The QEC array is constructed using experimentally-demonstrated quantum circuit elements: Josephson junctions, capacitors and $0 - \pi$ qubits[11–17]. The proposed QEC array departs from the established paradigm of probing confinement in condensed matter systems and starts with lattice quantum rotors. These lattice regularizations are particularly suitable for simulating a large class of strongly-interacting bosonic QFTs[18] due to rapid convergence to the scaling limit. While this was numerically observed in the semi-classical regime of the sine-Gordon (sG) model[19], here we

show that QECs are suitable for regularizing a strong-interacting, non-integrable bosonic QFT.

With a specific choice of interactions that arise naturally in QEC systems due to tunneling of Cooper pairs and pairs of Cooper pairs, we verify that the long-wavelength properties of the QEC array are described by a perturbed sG (psG) model, the continuum characteristics of which have been analyzed using semi-classical and perturbative techniques[20–24]. The corresponding euclidean action is

$$\mathcal{A}_{\text{psG}} = \int d^2x \left[ \frac{1}{16\pi} (\partial_\nu \varphi)^2 + V(\varphi) \right], \tag{1}$$

where $V(\varphi) = -2\mu \cos(\beta\varphi) - 2\lambda \cos(\beta\varphi/2)$ and $\lambda, \mu, \beta$ are parameters (see Supplementary Note I). Due to the presence of the perturbation $\propto \lambda$, the solitons and the antisolitons of the sG model experience a confining potential that grows linearly with their separation. This leads to the formation of mesonic excitations, analogous to the confinement phenomena occurring in the Ising model with a longitudinal field[25–30]. In the psG case, the free Ising domain-walls are replaced by interacting sG solitons. While predicted using semi-classical and perturbative analysis[22–24], quantitative investigations of confinement, direct evidence of the psG mesons and an experimentally-feasible proposal to realize this model have remained elusive so far. This is performed in this work.

[1]Department of Physics and Astronomy, Rutgers University, Piscataway, NJ 08854-8019, USA. ✉e-mail: ananda.roy@physics.rutgers.edu; sergeil@physics.rutgers.edu

## Results

Each unit cell of the one-dimensional QEC array [gray rectangle in Fig. 1] contains: (i) a Josephson junction on the horizontal link with junction energy (capacitance) $E_J(C_J)$, (ii) a parallel circuit of an ordinary Josephson junction [junction energy (capacitance) $E_{J_1}(C_1)$] and a $0 - \pi$ qubit[11-14] on the vertical link. The $0 - \pi$ qubit is realized using two Josephson junctions [junction energies (capacitances) $E_J'(C_J')$], together with two inductors with inductances $L$ [Fig. 1b]. In the limit $(L/C_J')^{1/2} \gg \hbar/(2e)^2$, this circuit configuration realizes a $\cos(2\phi)$ Josephson junction[14]. In the limit $C_J \gg C_{\text{eff}}$, where $C_{\text{eff}} = C_1 + C_2$, the QEC array is described by the Hamiltonian:

$$H = E_c \sum_{k=1}^{L} n_k^2 + \epsilon E_c \sum_{k=1}^{L} n_k n_{k+1} - E_J \sum_{k=1}^{L} \cos(\phi_k - \phi_{k+1})$$
$$- E_g \sum_{k=1}^{L} n_k - \sum_{a=1,2} E_{J_a} \sum_{k=1}^{L} \cos(a\phi_k), \qquad (2)$$

where $E_c = (2e)^2/2C_{\text{eff}}$ and we have chosen periodic boundary conditions. Here, $n_k$ is the excess number of Cooper pairs on each superconducting island and $\phi_k$ is the superconducting phase at each node, satisfying $[n_j, e^{\pm i\phi_k}] = \pm \hbar \delta_{jk} e^{\pm i\phi_k}$, with $\hbar$ set to 1 in the computations. Note that the eigenvalues of $n_k$-s can be both positive and negative integers, the latter corresponding to creation of holes in the superconducting condensate on the $k^{\text{th}}$ island. We approximate the exponentially-decaying, long-range interaction due to the capacitance $C_J$[31] with a nearest-neighbor interaction[32] of the form $\epsilon n_k n_{k+1}$, where the constant $\epsilon$ is $< 1$. Note that the confinement phenomena investigated here would continue to exist in the case $\epsilon = 0$. The third and fourth terms in Eq. (2) arise due to the coherent tunneling of Cooper-pairs between nearest-neighboring islands and due to a gate-voltage at each node. The last two cosine potentials of Eq. (2) respectively arise from tunneling of Cooper-pairs and pairs of

Cooper-pairs through the Josephson junction and the $0 - \pi$ qubit on the vertical link.

For $E_{J_2} = E_{J_1} = 0$, $H$ corresponds to a variation of the Hamiltonian of the Bose–Hubbard model[33,34] and conserves the total number of Cooper-pairs. As $E_J/E_c$ is increased from 0, the QEC array transitions from an insulating to superconducting phase. We focus on the superconducting phase obtained by increasing $E_J/E_c$ at constant density[35,36]. In the latter phase, the long-wavelength properties of the array are described by the free, compactified boson QFT[31,32], characterized by the algebraic decay of the correlation function of the lattice vertex operator: $\langle e^{i\phi_j} e^{-i\phi_k} \rangle \propto |j - k|^{-K/2}$, where $K$ is the Luttinger parameter. This algebraic dependence is verified in Fig. 2a by computing the corresponding correlation function using the density matrix renormalization group (DMRG) technique (The DMRG computations of this work were performed using the TeNPy package[37]). For the parameters in this work, the Luttinger parameter varies between $0 \leq K \leq 2$[32,38]. We further compute the dimensionless "Fermi/plasmon velocity", $u$, in the QEC array by analyzing the ground-state energy of the array with system-size (see Supplementary Note III) [Fig. 2c].

For $E_{J_2} \neq 0, E_{J_1} = 0$, keeping $E_J > E_c$, the QEC array realizes the sG model[19]. Now, the lattice model has a conserved $\mathbb{Z}_2$ symmetry, associated with the parity operator for the number of Cooper-pairs: $P = \prod_{k=1}^{L} e^{i\pi n_k}$. This symmetry leads to a two-fold degenerate ground state for this realization of the sG model. This is in contrast to the usual continuum formulation of the latter, where the ground state is one of the infinitely many vacua. The two degenerate states correspond to $\phi_k = 0$ and $\phi_k = \pi, k = 1, ..., L$, with the sG solitons and antisolitons interpolating between them. The sG coupling, $\beta$, is given by: $\beta = \sqrt{K/2} \in (0,1)$ (see Supplementary Note I).

We verify the sG limit of the QEC array as follows. First, we compute the scaling of the lattice operator $e^{i\phi_k}$, which, in the continuum limit, correspond to the vertex operator $e^{i\beta\varphi/2}$. The scaling with the coupling $E_{J_2}/E_c$ [Fig. 2b] yields the value of the sG coupling $\beta^2$ [Fig. 2c].

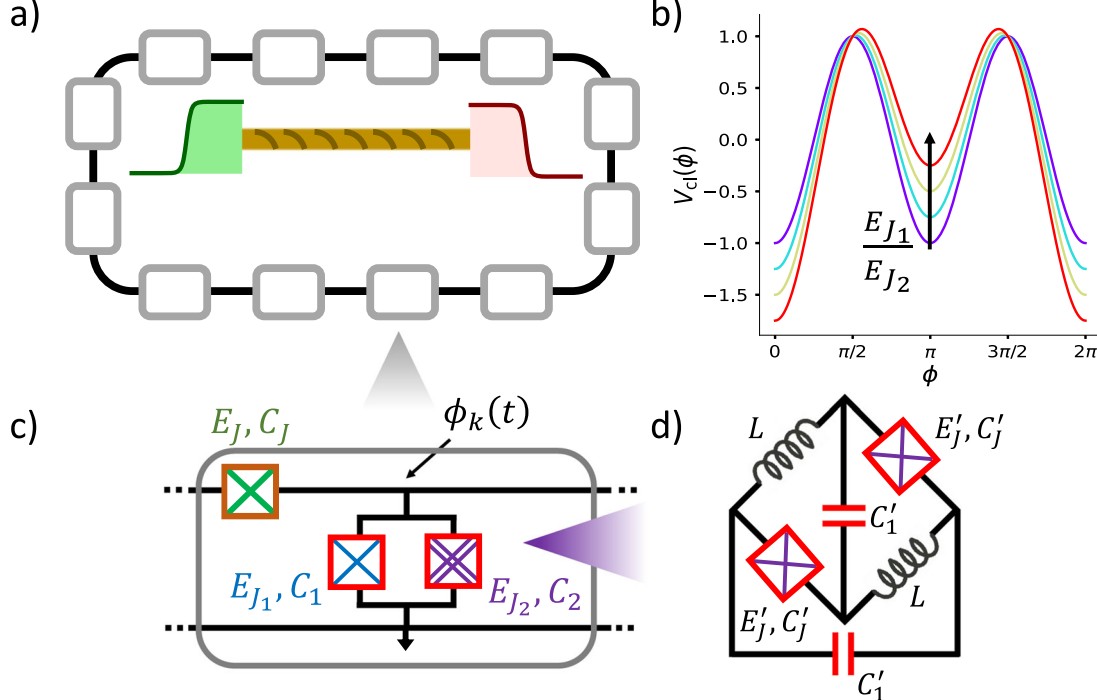

**Fig. 1 | Schematic of the QEC array.** Each unit cell (gray rectangle) of the QEC array (**a**) contains a Josephson junction (green cross) on the horizontal link. The vertical link (**c**) of the same contains a parallel circuit of an ordinary Josephson junction (blue cross) and a $\cos(2\phi)$ Josephson junction (purple crosses). The latter is formed by two Josephson junctions, two capacitors and two inductors (**d**)[14]. The variation of the classical potential, $V_{\text{cl}}$, [Eq. (1)] as $E_{J_1}/E_{J_2}$ increases from 0 in steps of 1/4 is shown in (**b**). For nonzero $E_{J_1}/E_{J_2}$, the solitons (green wavepacket) and antisolitons (maroon wavepacket), interpolating between the potential minima at $\phi = 0$ and $\phi = \pi$, experience a confining potential (yellow string in **a**), leading to the formation of mesonic bound states.

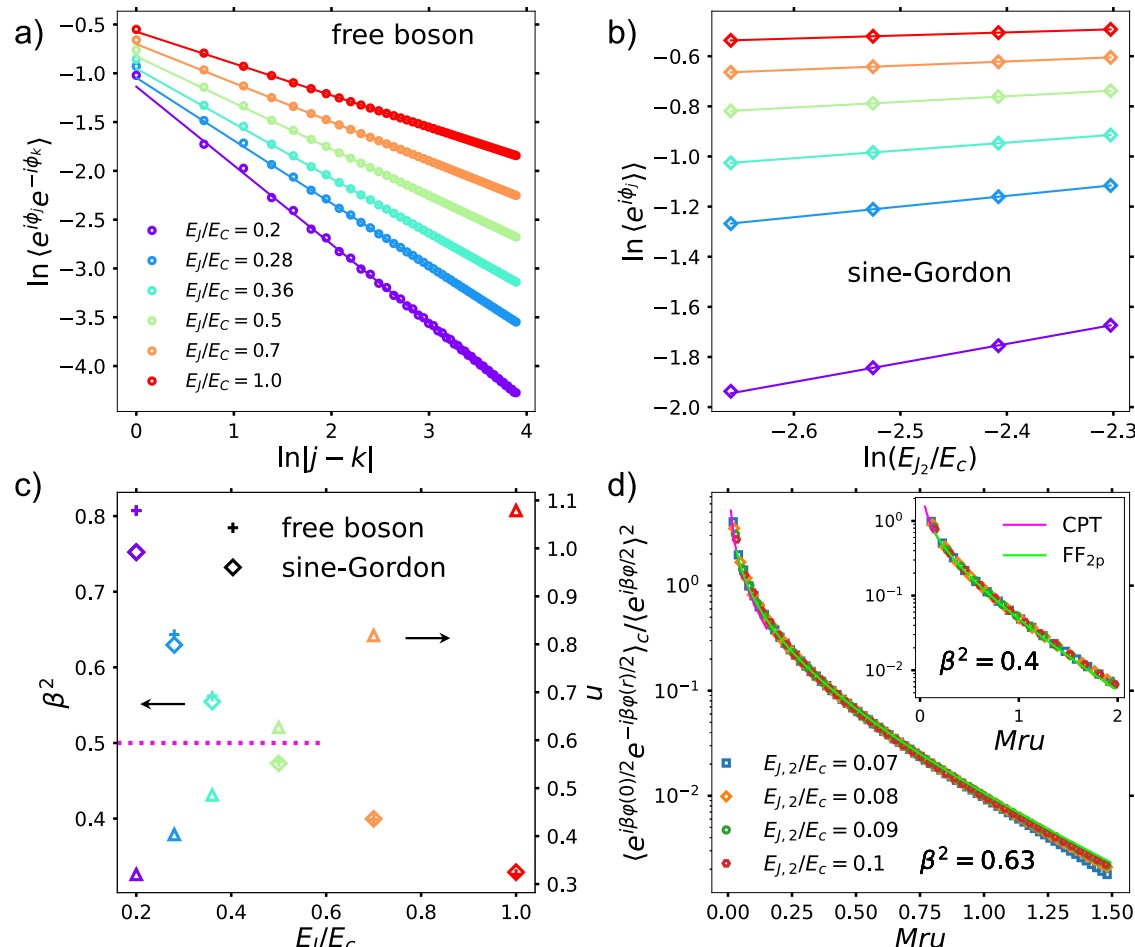

**Fig. 2 | DMRG results and comparison with analytical predictions. a** Verification of the power-law decay of the correlation functions of the lattice vertex operators for the free boson model obtained for $E_{J_1} = E_{J_2} = 0$ keeping $E_J/E_C$ finite. The obtained Luttinger parameter ($K = 2\beta^2$) from the slopes are plotted as pluses in **c**. **b** Scaling of the vertex operator expectation value with $E_{J_2}/E_C$ for the sG model. The values of the sG coupling obtained from this scaling are plotted as diamonds in **c**. The discrepancy between the sG result and the free-boson prediction as $\beta^2 \to 1$ occur due to corrections to scaling arising from the Kosterlitz–Thouless phase-transition occurring at $\beta^2 = 1$. The (dimensionless) Fermi/plasmon velocity, $u$, was obtained from the Casimir energy computation of the free theory (see Supplementary Note III). The free-fermion point of the sG model is indicated by the dotted magenta line. **d** Comparison of the normalized, connected two-point correlation function of the vertex operator $e^{i\phi_j} \sim e^{i\beta\phi/2}$ computed using DMRG and analytical computations in the repulsive ($\beta^2 \approx 0.63$) and the attractive ($\beta^2 \approx 0.4$, inset) regimes of the sG model. The ratio $1/Mu$, $M$ being the soliton-mass, was obtained by computing the correlation length from the infinite DMRG computation.

These values are compared with those expected from the free-boson computations. The discrepancy between the obtained values of $\beta^2$ for the sG and the free boson computations as $\beta^2 \to 1$ arises due to the Kosterlitz–Thouless phase-transition. We also compute the connected, two-point correlation function: $\langle e^{i\phi_j} e^{-i\phi_k} \rangle - \langle e^{i\phi_j} \rangle^2$. When normalized by $\langle e^{i\phi_j} \rangle^2$, the latter is given by a universal function, computable using analytical techniques. We compare the DMRG results with analytical predictions. We chose two representative values of $\beta^2$ to demonstrate the robustness of our results in both the attractive and repulsive regimes. The quantity, $Mu$, where $M$ is the soliton mass, is obtained numerically by computing the correlation length of the lattice model using the infinite DMRG technique. The short (long) distance behavior of the normalized, connected correlation function was computed using conformal perturbation theory (form-factors[39,40] computed by including up to two-particle contributions) (see Supplementary Note I). The results are shown as pink (lime) solid curves labeled CPT (FF$_{2p}$) in Fig. 2d.

The soliton-creating operators for the sG model[41,42] are defined on the lattice as: $O_s^q(k) = e^{2is\phi_k} \prod_{j<k} e^{-iq\pi n_j}$, where $q$ and $s$ are the topological charge and the Lorentz spin of the excitations. The current QEC incarnation of the sG model gives access to solitons with $s \in \{0, 1/2, 1\}$

and $q = \pm 1$. For definiteness, we consider $s = 0$. Figure 3a (empty markers) shows the energy cost, $T$, of separating a soliton-antisoliton pair, after they are created by application of $O_s^q$ at two different locations for different values of $\beta^2$. For the sG model, as expected, $T = 0$ for all values of the separation $d$. The corresponding phase-profile can be inferred by computing $\langle e^{i\phi_k} \rangle$ for different lattice sites, after normalizing with respect to the ground-state results [Fig. 3b].

The situation changes dramatically for the psG model, realized by making $E_{J_1} \neq 0$ in Eq. (2), while choosing the rest of the parameters as for the sG model. Due to the perturbing potential $\sim \cos(\phi_k)$, the sG solitons and the antisolitons experience a strong-confining potential energy, qualitatively similar to that experienced by the free, Ising domain walls under a longitudinal field[25–28]. We compute the energy-cost of separation $T$ for the psG model as in the sG case [Fig. 3a, filled markers]. The energy-cost grows proportional to the distance of separation: $T = \sigma d$, where $\sigma$ is the string-tension. The latter is numerically obtained by fitting to this linear dependence and shown as a function of $\beta$ in Fig. 3c. To leading order, $\sigma = 2\langle e^{i\phi_k} \rangle E_{J_1}/E_C$, where the expectation value $\langle e^{i\phi_k} \rangle$ is computed for the ground state of $H$ with $E_{J_1} = 0$. The discrepancy between the leading-order prediction and the numerical results for $\beta^2 \approx 0.736$ is due to the proximity to the

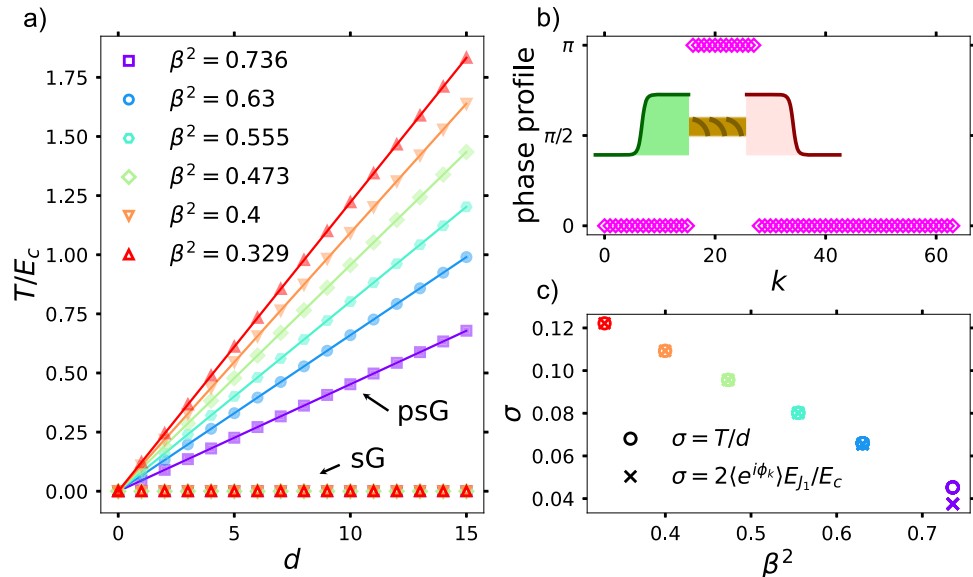

**Fig. 3 | DMRG results for the string tension for different choices of $\beta^2$, chosen by fixing $E_J/E_c$ [Fig. 2c], for $L$ = 64. a** The results are shown for $E_{J_2}/E_c$ = 0.1 for both the sG and psG models, while for the latter, $E_{J_1}/E_c$ = 0.1. Similar results were obtained for other choices. For the sG model (empty markers), after creating the soliton-antisoliton pair, there is no associated energy cost of separation. However, for the psG model (filled markers), due to the existence of the perturbing cosine potential $\propto E_{J_1}$ [Eq. (2)], the soliton and the antisoliton experience a confining force. This leads to an energy cost ($T/E_c$) growing linearly with separation $d$. **b** The corresponding phase-profile computed by creating a soliton-antisoliton pair and separating them by 12 lattice sites. **c** The corresponding string tension, $\sigma = T/d$ (empty circles) obtained from a linear fit of the data in **a**. The corresponding leading-order analytical predictions for $\sigma$ are denoted by crosses. The discrepancy between the predicted and obtained string-tension for $\beta^2 \approx 0.736$ occurs due to the proximity to the Kosterlitz–Thouless point ($\beta^2 = 1$).

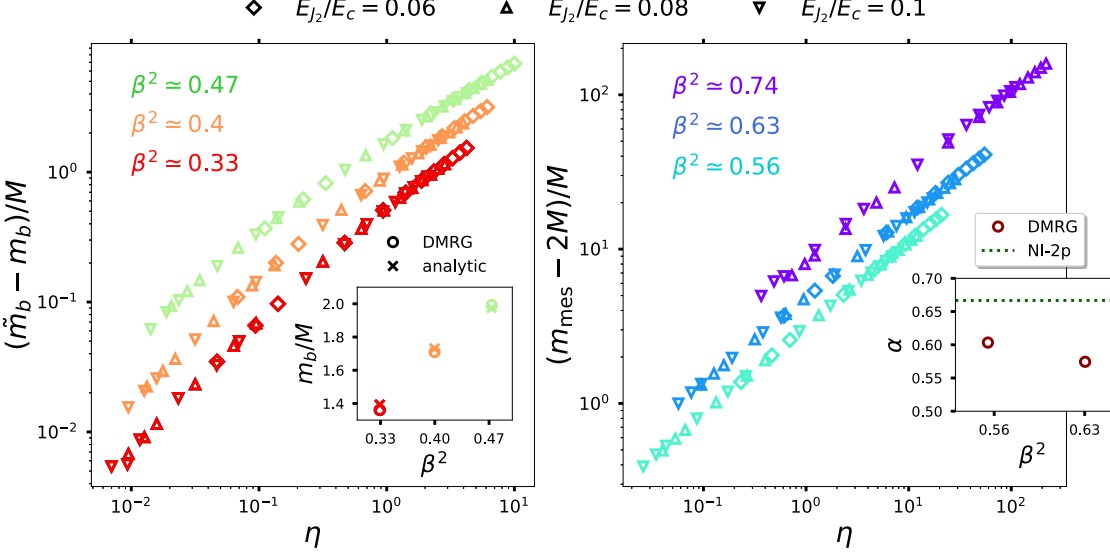

**Fig. 4 | DMRG results for the mass of the lightest particle of the psG model for $\beta^2 < 1/2$ (left) and $\beta^2 > 1/2$ (right), as a function of the dimensionless quantity $\eta$.** Here, $M(m_b)$ is the mass of the soliton (lightest breather) of the unperturbed sG model. The diamonds and triangles correspond to different choices of $E_{J_2}/E_c$. For small $\eta$, the lightest particle is the lightest sG breather (psG meson) for $\beta^2 < (>)1/2$. Using linear fit (see Supplementary Note II) of the numerical data for $\eta \ll 1$, we obtain the ratio $m_b/M$ (comparison with the analytical prediction in the left inset). The scaling of the psG meson mass is given by: $(m_{\text{mes}} - 2M)/M \sim \eta^\alpha$ for $\eta \ll 1$. The inset in the right panel shows the comparison of the $\alpha$ obtained using DMRG (circles) and those using non-interacting two-particle (NI-2p) approximation (dotted line).

Kosterlitz-Thouless point. The decrease of the string-tension with increasing $\beta^2$ can be viewed as a consequence of the increasing repulsion between the sG solitons and antisolitons with increasing $\beta^2$.

The spectrum of the psG model contains the newly-formed mesons and the charge-neutral sG breathers. The latter occur only for $\beta^2 < 1/2$ with their masses acquiring corrections due to the perturbing potential. Figure 4 shows DMRG results for mass of the lightest particle as a function of the dimensionless parameter $\eta = [E_{J_1}/E_c]/[E_{J_2}/E_c]^\nu, \nu = (1 - \beta^2/4)/(1 - \beta^2)$, for different choices of $E_{J_2}/E_c$. For small $\eta$, the psG mesons are heavier (with masses > 2$M$) than the breathers (with masses < 2$M$). We compute the mass of the lightest sG breather (psG meson) for $\beta^2 < (>)1/2$ from computation of the correlation lengths using infinite DMRG technique. For $\eta \ll 1$, the correction to the lightest breather mass can be expanded in powers

of $\eta$. We show a comparison of the obtained ratio $m_b/M$, $m_b$ being the lightest sG breather mass for $\eta = 0$, with the analytical predictions in the left inset. For a comparison of our numerical data with perturbative computation[23], see Supplementary Note IIB. For $\beta^2 > 1/2$, the spectrum contains only the psG mesons. The dependence of lowest psG meson mass is shown in Fig. 4 (right). For $\eta \ll 1$, a non-interacting two-particle (NI-2p) computation (see Supplementary Note IIC) predicts $(m_{mes} - 2M)/M \sim \eta^\alpha$, where $\alpha = \frac{2}{3}$. Comparison of the numerical results with the NI-2p computation is shown in the right inset. A more complete computation using the Bethe-Salpeter equation for the psG model is beyond the scope of this work.

## Discussion

To summarize, we have numerically demonstrated the confinement of sG solitons into mesonic bound states in a QEC array. We computed the associated string tension and computed the scaling properties of the mass of the lightest particle. In contrast to quantum spin-chains, which have been the defacto standard for lattice simulation of strongly-interacting QFTs, this work demonstrates the robustness and versatility of QEC to achieve this goal. Given that the primitive circuit elements of the proposed scheme have already been demonstrated, it is conceivable that predictions for additional physical properties of the psG model could be obtained using analog quantum simulation[43] in an experimental realization. For instance, a quench experiment would be able to capture signatures of the excitations with energy higher than what could reliably probed using DMRG. Consider the case when the junction energies of the blue Josephson junctions, $E_{J_1}$, in Fig. 1 are tunable. This can be accomplished by replacing the corresponding junctions by a SQUID loop with a magnetic flux threading the latter[44]. After preparing the system in the ground state of $H$ with $E_{J_1} = 0$, the coupling $E_{J_1}$ is turned on by tuning magnetic flux. Signatures of the confinement of the sG solitons can be obtained by probing the spectrum and the current-current correlation functions. Note that imperfections in an experimental realization of the $0 - \pi$ qubit that lead to an additional $\cos\phi$ potential would renormalize the coupling $E_{J_1}$ of Eq. (2) and does not pose an impediment towards investigation of the confinement phenomena analyzed in this work. Given the progress in the fabrication and investigation of large QEC arrays[45–47], we are optimistic of experimental vindication of our work.

The proposed QEC provides a starting point for the realization of a large number of one-dimensional QFTs. First, replacing the blue Josephson junction on the vertical link in Fig. 1 by a linear inductor gives rise to the renowned massive Schwinger model. Second, tuning a magnetic flux between the Josephson junction and the $0 - \pi$ qubit in each cell changes the perturbing potential in Eq. (2) from $\cos(\phi_k)$ to $\sin(\phi_k)$. For certain values of $E_{J_1}/E_{J_2}$, this induces a renormalization group flow from the gapped perturbed sine-Gordon model to a quantum critical point of Ising universality class[23,24,48]. Third, QECs provide a robust avenue to realize sG models with $a$-fold degenerate minima, where $a \in \mathbb{Z}$ (see Supplementary Note IV). The corresponding $\cos(a\phi)$ circuit element can be constructed by recursively using the $\cos\phi$ and $\cos 2\phi$ circuit elements. Perturbations of these sG models lead to not only soliton confinement and false-vacuum decays[49,50] present in the $a = 2$ case, but also all unitary minimal conformal field theory models[48,51]. Controlled realization of the latter multicritical Ising models opens the door to numerical and experimental investigation of a wide range of impurity problems that have so far been elusive.

## Data availability

The data used in the manuscript is available from the authors upon request.

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

## Acknowledgements

The authors acknowledge discussions with Johannes Hauschild, Robert Konik, Marton Kormos, Hubert Saleur, Gabor Takacs, and Yicheng Tang. This work was supported by a grant (825876, TDN) from the Simons Foundation (AR) and by NSF-PHY-2210187 (SLL).

## Author contributions

A.R. and S.L.L. contributed equally to this work.

## Competing interests

The authors declare no competing interests.
