## [Peer Review File · Nature Communications]

REVIEWER COMMENTS

Reviewer #1 (Remarks to the Author):

Key results:

The authors numerically demonstrated the confinement of sine-Gordon solitons into mesonic bound states. The precise target model of simulation is a variant of the Bose-Hubbard model that is expected to be approximated by a quantum electronic circuit (QEC) array. The numerical simulation was done by the density matrix renormalization group (DMRG) method implemented by the TeNPy package.

Validity:

The numerical results seem very clean and reliable.

Significance:

Given that a large system of QEC arrays has been experimentally developed, quantum analog simulation of lattice quantum field theories such as the one studied in the manuscript is feasible.

The current work exemplifies an approach (e.g., application of particle creation operators) for simulating confinement phenomena in lattice quantum field theories.

Data and methodology:

The overall methodology seems sound.

I could not find the value of $E_{\{J_1\}}$ for the perturbed lattice sine-Gordon model used in FIG. 3. If it is indeed not given anywhere, it should be specified.

Analytical approach:

The current work focuses on numerical simulations. The analytical results are mostly taken from previous works.

Suggested improvements:

In the definition of soliton-creating operators, I believe that n and q are the same, in view of [41]. It would be better to use only one of them. In fact, q appears only through $e^{-iq\pi n_j}$. If n_j are integers, $q=+1$ and $q=-1$ define the same operator. Do the authors allow n_j to be non-integers?

Above and below Eq. (2), \hbar appears in an inequality and in the commutation relation. The authors may want to remove it or explain that it is set to 1 in other places.

Clarity and context:

The overall writing is clear.

It would be useful to refer to sections in the supplementary material when more explanations are given there.

References:

It seems that enough appropriate references are supplied.

Reviewer #2 (Remarks to the Author):

In this work, the authors investigate by numerical and analytical methods how correlations functions of a perturbed sine-Gordon model reflect some properties of solitons, motivated by experiments with quantum circuits.

I would like to discuss first the proposed design for a possible experimental platform using superconducting quantum circuits. Let me stress that the demonstration of 0 - π qubits is not as clear-cut as suggested by the authors, and in fact only a "softer" version has been

so far realized, that does not offer the wanted protection for fault-tolerant quantum manipulation.

This being said, the authors propose to pair up the 0- π unit, namely a perfect $\cos(2\phi)$ term, with in parallel a regular Josephson junction, namely the usual $\cos(\phi)$ term. This seems unnecessarily complicated in terms of design, because:

1/ the goal of the 0- π qubit is to get rid precisely of $\cos(\phi)$, so why take this effort, if it is brought back in the end?

2/ if the goal is to break the 2π periodicity of the regular $\cos(\phi)$ potential, there are maybe simpler ways, for instance using a super-inductor potential $E_L \phi^2$.

I would like the authors to consider if there are simpler alternative routes to help experimentalists.

I also point out that any Josephson junction with high transparency (beyond the tunnel limit) will be described by a non-cosine potential, given by a known series of harmonics $\cos(n\phi)$, and looking typically as a skewed cosine potential. This also breaks integrability of sG model. Would the main result of soliton confinement apply also in that case?

A more technical point concerns Hamiltonian (2). As mentioned, the inverse capacitance matrix decays

exponentially over some screening length. I am guessing that incorporating this realistic ingredient in DMRG calculations is not easy due to non-locality. But I am wondering why the authors kept the nearest neighbor term ϵ , as it is not discussed in the main text. There is some cryptic mention of its use made in the Supplementary material, and I would ask the authors to clarify why it is important and what impact could it bring to the whole analysis.

It would help the reader in the caption of Fig. 2a to indicate that the "free boson" model is $E_{j1}=E_{j2}=0$, keeping E_j finite and non-linear. Perhaps "lattice XY model" or "Josephson junction array" would be a more appropriate terminology, I don't see why this is free boson unless $E_j \gg E_c$.

I was confused by the term "Fermi velocity" (shown in Fig. 2c), do the authors rather talk about the renormalized plasmon velocity?

It is clear that the correlation function $\langle \exp(i\phi(r) - i\phi(0)) \rangle$ decays algebraically with distance r in the free boson case, Fig. 2a. In what way is the exponential behavior shown in Fig. 2d due to solitons, and why should one consider such specific form of connected correlation function, specifically

$$\langle e^{i\phi_j} e^{i\phi_k} \rangle - \langle e^{i\phi_j} \rangle^2$$

rather than

$$\langle e^{i\phi_j} e^{i\phi_k} \rangle - \langle e^{i\phi_j} \rangle \langle e^{i\phi_k} \rangle?$$

A minor question: why is there an upper index "n" in the soliton operator? Is there any connection of this operator to the usual Jordan Wigner string for spin systems? Maybe some physical motivation would be useful to have here.

Could the authors also indicate in the main text how the masses of the bound solitons pairs are computed in practice from DMRG (Fig 4)?

A final but important question is the following: is there really a simple experimental way to measure the correlation functions of the vertex operator? In simpler contexts (e.g. the Kondo screening cloud), this has not been achieved so far.

Reviewer #3 (Remarks to the Author):

The paper adds an additional example to the list of confinement phenomena in low-dimensional physics. This is done using language and methods borrowed by quantum field theory. In this context there is a simple way to understand the emergence of confinement of topological excitations (as is the case in this paper): confinement occurs each time that a perturbing field is non-local wrt the particle excitations. This universal explanation was discussed for the first time in the paper

G. Delfino, G. Mussardo, P. Simonetti, Non integrable quantum field theories as perturbations of certain integrable models, Nucl. Phys. B 473 (1996), 469.

See the discussion at pag 495, for the concrete example of the kink confinement in the low-temperature phase of the Ising model once perturbed by the magnetization operator. In that paper it was also developed the Form Factor Perturbation Theory (FFPT) which is a very efficient way to control the effect of perturbation, in alternative to the usual (Feynman like) power perturbation series.

Unfortunately the paper mentioned above is missing in the list of references as well as the important paper by McCoy and T.T. Wu (probably the seminal paper that firstly brought the attention of confinement in low-dimensional system) which is only listed en-passant in the Supplementary Materials but not in the main text.

Moreover, the mechanism which leads to confinement of topological excitations due to the non-local property of the perturbed operator was discussed in great detail in the paper

G. Delfino and G. Mussardo, Non-integrable aspects of the multi-frequency sine-Gordon model, Nucl. Phys. B 516 (1998), 675.

See, in particular Section 4 of this paper, whose explicit title is Non-locality, soliton confinement and phase transition.

The results discussed in the paper by Roy and Lukyanov seem then a simple consequence of the analysis done in the paper aforementioned on multi-frequency Sine-Gordon, given that the quantum electronic circuit is, in the continuum limit, just the double sine-gordon model. There are no new conceptual results which cannot be found in the paper mentioned above.

Even though the results presented in the paper by Roy and Lukyanov are sound, in light of the previous considerations the narrative of their paper on confinement in low dimensional system seems quite peculiar. To give few examples: the seminal paper on confinement by McCoy and Wu (dated 1978) is only quoted in the Supplementary Materials while the paper NPB516, published in 1998, on multi-frequency sine-gordon (which is the key subject of the manuscript) is only mentioned en-passant. On the other hand, the authors prefer to put as first papers of their list a serie of papers which were either inspired by those papers or in any case published much later.

Response to Reviewers: NCOMMS-23-09922A-Z

Ananda Roy and Sergei Lukyanov

Report of Reviewer #1

In the following we address the reviewer's comments/suggestions one by one. We reproduce the reviewer's comments in (blue) italics, with our responses in (black) roman. The explicit changes to the manuscript are highlighted as bullet points.

Key results:

The authors numerically demonstrated the confinement of sine-Gordon solitons into mesonic bound states. The precise target model of simulation is a variant of the Bose-Hubbard model that is expected to be approximated by a quantum electronic circuit (QEC) array. The numerical simulation was done by the density matrix renormalization group (DMRG) method implemented by the TeNPy package.

Validity:

The numerical results seem very clean and reliable.

We thank the reviewer for his/her positive evaluation of our numerical results.

Significance:

Given that a large system of QEC arrays has been experimentally developed, quantum analog simulation of lattice quantum field theories such as the one studied in the manuscript is feasible.

The current work exemplifies an approach (e.g., application of particle creation operators) for simulating confinement phenomena in lattice quantum field theories.

We thank the reviewer for his/her positive evaluation of our manuscript.

Data and methodology:

The overall methodology seems sound.

I could not find the value of $E_{\{J_1\}}$ for the perturbed lattice sine-Gordon model used in FIG. 3. If it is indeed not given anywhere, it should be specified.

We thank the referee for his/her positive evaluation of our methodology.

Indeed, the values of bare lattice couplings $E_{\{J_1\}}$, $E_{\{J_2\}}$ were not mentioned in the manuscript. We have remedied this in the current version of the manuscript.

- *Caption in Fig. 3: The results are shown for $E_{J_2}/E_c = 0.1$ for both the sG and psG models, while for the latter, $E_{J_1}/E_c = 0.1$. Similar results were obtained for other choices.*

Analytical approach:

The current work focuses on numerical simulations. The analytical results are mostly taken from previous works.

Indeed, the focus of the current work is to demonstrate the feasibility of simulating perturbed sine-Gordon models with quantum electronic circuits. Since the lattice model is non-integrable, the relevant lattice computations were done using a numerical technique. However, in addition to the numerical computations, a new analytical computation was performed for the calculation of the sine-Gordon correlation functions in the short-distance limit. The results of the new analytical computation are shown as magenta curves labeled in CPT in Fig. 2. The corresponding analytical predictions are given in Sec. IB of the Supplementary Material.

Suggested improvements:

In the definition of soliton-creating operators, I believe that n and q are the same, in view of [41]. It would be better to use only one of them. In fact, q appears only through $e^{-iq \sum_j n_j}$. If n_j are integers, $q=+1$ and $q=-1$ define the same operator. Do the authors allow n_j to be non-integers?

Indeed, the superscript should be O_s^q and not O_s^n . We have corrected this in the updated version of the manuscript.

Furthermore, both $q = \pm 1$ define the same operator since the n_j -s are integers. The latter are restricted to be integers since they correspond to the excess number of Cooper-pairs or holes in the superconducting island.

- *Paragraph 3 on page 3: ... $O_s^q = \dots$*
- *Footnote 32: Note that the eigenvalues of n_k -s can be both positive and negative integers, the latter corresponding to creation of holes in the superconducting condensate on the k^{th} island.*

Above and below Eq. (2), \hbar appears in an inequality and in the commutation relation. The authors may want to remove it or explain that it is set to 1 in other places.

We have remedied this in the current version of the manuscript.

- *Below Eq. 2: ...with \hbar set to 1 in the computations.*

Clarity and context:

The overall writing is clear. It would be useful to refer to sections in the supplementary material when more explanations are given there.

We thank the reviewer again for his/her positive evaluation of our writing. We have added the specific sections while citing the supplementary materials.

- *Below Fig. 2: Ref. [25], Sec. III*
- *Top paragraph of page 3: Ref. [25], Sec. I*
- *Second paragraph on page 3: Ref. [25], Sec. IB*
- *First paragraph on page 4: see Sec. IIB of Ref. [25]. ... Ref. [25], Sec. IIC ...*

References:

It seems that enough appropriate references are supplied.

We thank the reviewer again for his/her positive evaluation of our work.

Report of Reviewer #2

In the following we address the reviewer's comments/suggestions one by one. We reproduce the reviewer's comments in (blue) italics, with our responses in (black) roman. The explicit changes to the manuscript are highlighted as bullet points.

In this work, the authors investigate by numerical and analytical methods how correlations functions of a perturbed sine-Gordon model reflect some properties of solitons, motivated by experiments with quantum circuits.

We thank the reviewer for his/her reading of our manuscript.

I would like to discuss first the proposed design for a possible experimental platform using superconducting quantum circuits. Let me stress that the demonstration of 0- π qubits is not as clear-cut as suggested by the authors, and in fact only a "softer" version has been so far realized, that does not offer the wanted protection for fault-tolerant quantum manipulation.

Indeed, the experimental works (Refs. [14-16]) have realized "softer" versions of the $\cos 2\phi$ potential with contamination coming from $\cos \phi$ term. While this is a problem for fault-tolerant quantum manipulation of $0 - \pi$ qubits, it is the opposite for the realization of the analyzed perturbed sine-Gordon model. The presence of a spurious $\cos \phi$ term in the realized $0 - \pi$ qubit would only trivially renormalize the bare lattice coupling E_{J_1} , leaving the confinement phenomena investigated in the perturbed sine-Gordon model unchanged. We emphasize again that this potential experimental imperfection raises a problem in realization of the *unperturbed* sine-Gordon model, but only trivially renormalizes the lattice coupling of the perturbed sine-Gordon model.

- *Footnote 48:* Note that imperfections in an experimental realization of the $0 - \pi$ qubit that lead to an additional $\cos \phi$ potential would renormalize the coupling E_{J_1} of Eq. (2) and does not pose an impediment towards investigation of the confinement phenomena analyzed in this work.

This being said, the authors propose to pair up the 0- π unit, namely a perfect $\cos(2\phi)$ term, with in parallel a regular Josephson junction, namely the usual $\cos(\phi)$ term. This seems unnecessarily complicated in terms of design, because:

1/ the goal of the 0- π qubit is to get rid precisely of $\cos(\phi)$, so why take this effort, if it is brought back in the end?

The $\cos 2\phi$ potential is used to realize the unperturbed sine-Gordon model. In the unperturbed model, the solitons and anti-solitons experience no confining force (see Fig. 3 of the main text). The perturbation in the form of the $\cos \phi$ potential gives rise to the confining force that leads to the formation of the mesonic bound state. The ordinary Josephson junction and the corresponding $\cos \phi$ potential is brought back in the end to provide a *tunable* confining potential. As explained above, imperfections in an experimental realization leading of an addition $\cos \phi$ potential will renormalize the bare lattice coupling, leaving the essential physics analyzed in the manuscript unchanged.

2/ if the goal is to break the 2- π periodicity of the regular $\cos(\phi)$ potential, there are maybe simpler ways, for instance using a super-inductor potential $E_L \phi^2$.

I would like the authors to consider if there are simpler alternative routes to help experimentalists.

The referee is correct that there are additional ways to break the periodicity of the $\cos 2\phi$ potential. In fact, the mentioned quadratic perturbing potential will give rise to the well-known massive Schwinger model. The latter, while different from the perturbed sine-Gordon model in many aspects, also exhibits confinement phenomena similar to what is analyzed in this work. This point has also already been mentioned in one of the authors works (Sec. 6 of Ref. 19).

However, this is not the model analyzed in this work. As explained in the manuscript, the current work uses *experimentally demonstrated circuit elements*. As such, we do not see any particular reason to favor the model proposed by the reviewer over the model analyzed in this work.

Last, but not least, the current approach of perturbing the sine-Gordon model by one/several harmonic term(s) gives rise to a richer set of physical phenomena compared to the quadratic perturbation suggested by the reviewer. This has been clarified in the conclusion of the updated manuscript. For example, by tuning the relative phase, one arrives at a perturbation of the form $\sin \phi$, which gives rise to an Ising type phase-transition. Recently, one of the authors demonstrated that all unitary minimal models of two-dimensional conformal field theories can be realized using this approach (arXiv:2306.04346).

- *Penultimate paragraph of main text:* The proposed QEC provides a starting point for the realization of a large number of one-dimensional QFTs. First, replacing the blue Josephson junction on the vertical link in Fig. 1 by a linear inductor gives rise to the renowned massive Schwinger model. Second, tuning a magnetic flux between the Josephson junction and the $0 - \pi$ qubit in each cell changes the perturbing potential in Eq. (2) from $\cos \phi_k$ to $\sin \phi_k$. For certain values of E_{J_1}/E_{J_2} , this induces a renormalization group flow from the gapped perturbed sine-Gordon model to a quantum critical point of Ising universality class [21, 22, 48].
- *Sec. IV of Supplement:* see Supplementary Material, Sec. IV.

I also point out that any Josephson junction with high transparency (beyond the tunnel limit) will be described by a non-cosine potential, given by a known series of harmonics $\cos(n\phi)$, and looking typically as a skewed cosine potential. This also breaks integrability of sG model. Would the main result of soliton confinement apply also in that case?*

Indeed, starting with a potential $\cos n\phi$ would give rise to a sine-Gordon model with n -fold degenerate vacua with the solitons and antisolitons interpolating between them. Perturbations of the form $\cos m\phi$ with $m < n$ can rise to the confining potential for the solitons and antisolitons. The simplest generalization is the case $n = 4, m = 2$. We analyzed this case extensively, but decided to present the results for the case $n = 2, m = 1$ since the $\cos 4\phi$ circuit element, while straightforward to construct using existing $\cos 2\phi$ elements, has not been experimental demonstrated so far.

In fact, the situation for the case of perturbations of the $\cos n\phi$ potential is even more exciting. By tuning strengths of the perturbations, minima of the starting sine-Gordon model can be made to successively coalesce giving rise to one-dimensional multicritical quantum Ising models, equivalently, the diagonal unitary models of conformal field theories. Explicit demonstrations of this fact has been presented in A. Roy, arXiv:2306.04346 with numerical signatures of the Ising and tricritical Ising models. Controlled realization of different universality classes of quantum critical models is highly nontrivial

coveted and can give rise to numerical and experimental probing of a wide range of non-perturbative phenomena in quantum field theories.

- *Penultimate paragraph of main text:* Third, QECs provide a robust avenue to realize sG models with a -fold degenerate minima, where $a \in \mathbb{Z}$. The corresponding $\cos(a\phi)$ circuit element can be constructed by recursively using the $\cos \phi$ and $\cos 2\phi$ circuit elements considered in this work. Perturbations of these sG models lead to not only soliton confinement and false-vacuum decays [53, 54] present in the $a = 2$ case, but also all unitary minimal conformal field theory models [52, 55]. Controlled realization of the latter multicritical Ising models opens the door to numerical and experimental investigation of a wide range of impurity problems that have so far been elusive.
- *Sec. IV of Supplement:* see Supplementary Material, Sec. IV.

A more technical point concerns Hamiltonian (2). As mentioned, the inverse capacitance matrix decays exponentially over some screening length. I am guessing that incorporating this realistic ingredient in DMRG calculations is not easy due to non-locality. But I am wondering why the authors kept the nearest neighbor term epsilon, as it is not discussed in the main text. There is some cryptic mention of its use made in the Supplementary material, and I would ask the authors to clarify why it is important and what impact could it bring to the whole analysis.

Indeed, as the reviewer notes, including an exponentially decaying interaction is possible, albeit complicated, in a DMRG computation. Our goal of keeping the nearest neighbor term was to be closer to systems that can be realized in practice, as considered, for example, in Ref. 34. The ratio of the capacitance to ground (C_g) and the junction capacitance (C_j) can be tuned in an experimental setup such that beyond-nearest-neighbor interactions can be safely neglected.

The parameter ϵ plays no profound role in our analysis and the physical phenomena analyzed in this work would continue to exist for the case $\epsilon = 0$. In the current QEC model, the sine-Gordon coupling is related to the Luttinger parameter of the free theory as: $\beta^2 = K/2$. Having a nearest-neighbor interaction of the form $n_j n_k$ in addition to the onsite terms n_j^2 leads to $K \in [0, 2]$, allowing investigation of both attractive ($\beta^2 < 1/2$) and repulsive ($\beta^2 > 1/2$) regimes of the sine-Gordon model. With only onsite terms, the Luttinger parameter is restricted to $[0, 1]$, which restricts the sine-Gordon model to be only in the attractive regime. However, the confinement of sine-Gordon solitons into mesonic excitations continue to happen in this parameter regime as is also demonstrated in this work (see Fig. 4, left).

Before leaving this topic, we remark that both the attractive and repulsive regimes of the sine-Gordon model with perturbation can be analyzed in the $\epsilon = 0$ case by considering a potential of the form $= -E_{J_4} \cos 4\phi_k - E_{J_2} \cos 2\phi_k$. We have explained this in a new section (Sec. IV) in the supplementary materials.

- *Footnote 35:* The confinement phenomena described in this manuscript would continue to exist in the case $\epsilon = 0$.
- *Sec. IV of Supplementary Materials:* see Sec. IV of Supplementary Materials.

It would help the reader in the caption of Fig. 2a to indicate that the "free boson" model is $E_{J1}=E_{J2}=0$, keeping E_J finite and non-linear. Perhaps "lattice XY model" or "Josephson junction array" would be a more appropriate terminology, I don't see why this is free boson unless $E_J \gg E_c$.

Indeed, the computation of Fig. 2a is performed for $E_{J_1} = E_{J_2} = 0$, keeping E_J/E_c finite. We have added a phrase to that end.

We point out to the reviewer that the intuition that the free-boson phase arises for $E_J \gg E_c$ is based on a semi-classical analysis and is *not quantitatively correct*. As shown in Ref. 39, Appendix B (see also Sec. I of Supplementary Materials), the Kosterlitz-Thouless transition at constant density for half-filling occurs at $\frac{E_J}{E_c} \sim 0.1$. We consider the cases $\frac{E_J}{E_c} \geq 0.2$ as the corrections to the scaling are the smallest for this regime. We draw the reviewer's attention to the near-perfect algebraic dependence obtained using DMRG simulations in Fig. 2a for $\frac{E_J}{E_c} = 0.2$.

- *Caption of Fig. 2: ...obtained for $E_{J_1} = E_{J_2} = 0$ keeping E_J/E_c finite*

I was confused by the term "Fermi velocity" (shown in Fig. 2c), do the authors rather talk about the renormalized plasmon velocity?

We do talk about what is also in the superconducting literature referred to as plasmon velocity. To that end, we have added the corresponding word.

- *Caption to Fig. 2 and in the paragraph below it: "Fermi/plasmon velocity"*

It is clear that the correlation function $\langle \exp(i\phi(r) - i\phi(0)) \rangle$ decays algebraically with distance r in the free boson case, Fig. 2a. In what way is the exponential behavior shown in Fig. 2d due to solitons, and why should one consider such specific form of connected correlation function, specifically $\langle e^{i\phi_j} e^{i\phi_k} \rangle - \langle e^{i\phi_j} \rangle \langle e^{i\phi_k} \rangle$ rather than $\langle e^{i\phi_j} \rangle^2$?

Firstly, as mentioned in paragraph 3, page 3, the numerical simulations for the correlation functions were done using the 'infinite DMRG' technique. In this method, one numerically simulates the ground state of a *translation-invariant system*. Thus, the two quantities $\langle e^{i\phi_j} \rangle \langle e^{i\phi_k} \rangle$ and $\langle e^{i\phi_j} \rangle^2$ are equal.

Secondly, the behavior of the two-point function for the sine-Gordon model is not a simple exponential (see Sec. IB of Supplementary Materials). In fact, that it is not a simple exponential is precisely the signature of the strongly interacting regime of the sine-Gordon model. The computation of the two-point function was performed using conformal perturbation theory and form-factor techniques. When using form-factors, we included the contribution of all the relevant particles (breathers, solitons and antisolitons) to the corresponding order. The techniques used in this work are standard and have been used in the integrability and QFT community since 1990s. A full elucidation of the techniques would be impossible in this manuscript. We point the reviewer to the computation in Sec. I of the Supplementary Material or standard textbooks such as Ref. 40.

A minor question: why is there an upper index "n" in the soliton operator? Is there any connection of this operator to the usual Jordan Wigner string for spin systems? Maybe some physical motivation would be useful to have here.

The upper index should be q which stands for the topological charge of the soliton, which in general, can be any integer. There was a misprint in the earlier version of the manuscript. We have corrected this in the current version.

- *Paragraph 3 on page 3: ... $O_s^q = \dots$*

Finally, we note that the non-local soliton operators for half-integer spins are fermionic (see Ref. 41, for instance). At the free-fermion point, they could be related to fermions arising from Jordan-Wigner transformation. However, since we do not specifically analyze the free-fermion point, we refrain from discussing this point which has been addressed in the literature before (see, e.g., Refs. 44, 45).

Could the authors also indicate in the main text how the masses of the bound solitons pairs are computed in practice from DMRG (Fig 4)?

The meson and breather masses in Fig. 4 were computed from the computation of the correlation length using infinite DMRG technique. Similar computation is done in Fig. 2 for the soliton mass. We have clarified this in the current version of the manuscript.

- *Bottom-right paragraph, page 3: from computation of the correlation lengths using infinite DMRG technique...*

A final but important question is the following: is there really a simple experimental way to measure the correlation functions of the vertex operator? In simpler contexts (e.g. the Kondo screening cloud), this has not been achieved so far.

Indeed, experimental measurement of correlation function of vertex operators is challenging using superconducting circuit technology. However, as in the case of the free-boson model, many universal features can be extracted using other observables. For the free-boson case, the Luttinger parameter, which could be obtained using the two-point correlation function, is obtained by introducing an impurity and measuring transport characteristics. For the sine-Gordon model, the two-point function has a much richer universal behavior (see Sec. I of Supplementary Materials) as opposed to the simple algebraic decay of the free-boson model. A full experimental characterization of this correlation function would be remarkable and nontrivial and lies outside the scope of this work. Nevertheless, as also mentioned in the manuscript, quantities such as the soliton/meson masses could be obtained using simpler spectroscopic probes.

Report of Reviewer #3

In the following we address the reviewer's comments/suggestions one by one. We reproduce the reviewer's comments in (blue) italics, with our responses in (black) roman. The explicit changes to the manuscript are highlighted as bullet points.

The paper adds an additional example to the list of confinement phenomena in low-dimensional physics. This is done using language and methods borrowed by quantum field theory. In this context there is a simple way to understand the emergence of confinement of topological excitations (as is the case in this paper): confinement occurs each time that a perturbing field is non-local wrt the particle excitations. This universal explanation was discussed for the first time in the paper

G. Delfino, G. Mussardo, P. Simonetti, Non integrable quantum field theories as perturbations of certain integrable models, Nucl. Phys. B 473 (1996), 469.

See the discussion at pag 495, for the concrete example of the kink confinement in the low-temperature phase of the Ising model once perturbed by the magnetization operator. In that paper it was also developed the Form Factor Perturbation Theory (FFPT) which is a very efficient way to control the effect of perturbation, in alternative to the usual (Feynman like) power perturbation series.

Unfortunately the paper mentioned above is missing in the list of references as well as the important paper by McCoy and T.T. Wu (probably the seminal paper that firstly brought the attention of confinement in low-dimensional system) which is only listed en-passant in the Supplementary Materials but not in the main text.

We thank the reviewer for his/her reading of our work and for pointing out this reference. We apologize for the accidental omission of this reference from our list. We have included this in the current version of the manuscript. We note that although working on related models such as the Ising model, the mentioned references does not analyze the model investigated in our work: the perturbed sine-Gordon model.

The work of McCoy-Wu is seminal in pointing out the existence of confinement in low-dimensional spin systems. We do cite it in the supplement where we use the results of McCoy and Wu. In hindsight, we agree with the reviewer that it should in fact be cited early in the main text. We have remedied this in the current version of the manuscript.

Finally, we are aware of the perturbative framework developed in the reference quoted by the reviewer. However, we are not sure what the reviewer suggests by "an efficient way to control the perturbation" when the goal is to simulate a strongly interacting model. The quoted framework is perturbative and as such, can be used to make perturbative predictions such as corrections to masses of particles already existing in the unperturbed model. In fact, we provide explicit comparison to the said computations in Sec. II of the Supplementary Material and demonstrate how existing computations fall short of explaining actual behavior of the breather masses in the strong coupling regime.

However, such a perturbative computation is powerless to make *quantitative* predictions for physical quantities such as the masses of mesons analyzed in this work. We were unable to find any quantitative

computation of the meson masses or string tension for the perturbed sine-Gordon model in 1996 and 1998 references quoted by the reviewer.

In contrast to earlier works on the perturbed sine-Gordon model, this work provides a *numerically-tractable, experimentally-realizable lattice model that can quantitatively investigate both the strong and weak coupling regimes of the perturbed sine-Gordon model*. While a perturbative framework may be valuable for obtaining predictions for certain observables, in our opinion, robust ab-initio lattice computations and proposals for experimental realizations of strongly interacting QFTs such as the one presented in this work remain invaluable for the advancement of the field of QFT, quantum simulation and condensed matter physics.

- Added Refs. 4 and 21 to the main text.
- *Top paragraph, page 2:* ... quantitative investigations of confinement, direct evidence of the psG mesons and an experimentally-feasible proposal to realize this model have remained elusive so far....

Moreover, the mechanism which leads to confinement of topological excitations due to the non-local property of the perturbed operator was discussed in great detail in the paper

G. Delfino and G. Mussardo, Non-integrable aspects of the multi-frequency sine-Gordon model, Nucl. Physics. B 516 (1998), 675.

See, in particular Section 4 of this paper, whose explicit title is Non-locality, soliton confinement and phase transition.

The results discussed in the paper by Roy and Lukyanov seem then a simple consequence of the analysis done in the paper aforementioned on multi-frequency Sine-Gordon, given that the quantum electronic circuit is, in the continuum limit, just the double sine-gordon model. There are no new conceptual results which cannot be found in the paper mentioned above.

We thank the reviewer for pointing out this reference (already present in the reference list, Ref. 22 in the updated manuscript). However, we respectfully disagree with statement that “there are no new conceptual results” and that the results discussed in this paper are “a simple consequence of the analysis” done in the 1996 and 1998 references quoted by the reviewer.

As the reviewer points out, that confinement of solitons occur in this perturbed sine-Gordon model has been noticed before. We do not claim otherwise in this manuscript. We point the reviewer to the last two sentences of the top paragraph of page 2, which read:

“While predicted using semi-classical and perturbative analysis [22– 24], quantitative investigations of confinement and any direct evidence of the psG mesons have remained elusive so far. This is performed in this work.”

The purpose of the manuscript is to show such a strongly-interacting QFT can be reliably probed in a *numerically-tractable lattice model based on quantum circuit elements that currently exist in the laboratory*. This is important for the following reasons.

First, the current one-dimensional QFT model and its relatives exhibit non-perturbative phenomena that can rarely be probed in a controlled setting in ab-initio lattice simulations or table-top experiments. In

this work, we point out that *current* superconducting circuit technology can reliably simulate the strong coupling regime of such a QFT. We provide convincing numerical evidence demonstrating the validity of our proposal. As such, this proposal advances the field of quantum simulation by demonstrating the possibility of simulation of a strongly interacting QFT with *existing* quantum circuit elements.

Second, in contrast to lattice computations based on quantum spin chains, which have been the gold-standard for simulations of a large class of low-dimensional QFTs, this work provides an alternate paradigm for their analysis using quantum circuits. This is particularly important since quantum circuits exhibit very little corrections to scaling. Remarkably, we demonstrate that even an array of *only 64* Josephson junctions (Fig. 3 in the manuscript) is sufficient to capture the confinement phenomena in the strong coupling regime of this model. Note that arrays of thousands of Josephson junctions are routinely realized in experimental groups across the world.

Third, quantum circuits enable realizations of perturbed sine-Gordon models with n -fold degenerate vacua with $n > 2$. We are not aware of any other schemes for systematic lattice realization of quantum sine-Gordon models, perturbed or otherwise, with more than two vacua. Generalizations of the XYZ chain-based realization of the sine-Gordon model to achieve this goal would lead to highly nonlocal interactions, which are difficult to simulate numerically and are not realizable experimentally. While in this work, we focused on the case $n = 2$ since the corresponding circuit elements have been realized in experiments, it is straightforward to generalize this scheme. This has been recently used in Ref. 52 to give rise to multicritical Ising models. We have now added an additional section on this topic in the current version of the supplementary material. As a simple demonstration of the power of this approach, we present the scaling curve for the sine-Gordon correlation functions for $\beta^2 \sim 0.92$ and $\beta^2 \sim 0.99$ in Fig. 5 of the supplementary material. We intend to report more on this case and the related Wess-Zumino-Witten model in an upcoming work.

In summary, our work advances the field of low-dimensional QFT, quantum simulation and condensed matter physics by considering the new paradigm of mesoscopic quantum circuit regularizations of QFTs. To that end, we investigate a numerically-tractable, experimentally-realizable lattice model. We again reiterate our belief that perturbative continuum computations do not invalidate the importance of realistic proposals for realization of QFTs in table-top experiments and demonstration of their viability using numerical techniques.

- *Bottom paragraph, page 4:* In contrast to quantum spin-chains which have been the defacto standard for lattice simulation of strongly-interacting QFTs, this work demonstrates the robustness and versatility of QEC to achieve this goal. Given that the primitive circuit elements of the proposed scheme have already been demonstrated, it is conceivable that predictions for additional physical properties of the psG model could be obtained using analog quantum simulation [46] in an experimental realization. For instance, ...
- *Sec. IV of Supplementary Materials:* see Sec. IV of Supplementary Materials.

Even though the results presented in the paper by Roy and Lukyanov are sound, in light of the previous considerations the narrative of their paper on confinement in low dimensional system seems quite peculiar. To give few examples: the seminal paper on confinement by McCoy and Wu (dated 1978) is only quoted in the Supplementary Materials while the paper NPB516, published in 1998, on multi-frequency sine-gordon (which is the key subject of the manuscript) is only mentioned en-passant. On the other

hand, the authors prefer to put as first papers of their list a serie of papers which were either inspired by those papers or in any case published much later.

We thank the reviewer for his/her assessment of our work as sound. We have updated the reference list. The first mention of confinement in spin chains comes with the reference to McCoy and Wu's work and the first mention of the perturbed sine-Gordon model comes with several references which include both references mention by the reviewer.

- Added Refs. 4 and 21 to the main text.

REVIEWERS' COMMENTS

Reviewer #1 (Remarks to the Author):

The authors adequately addressed the issues raised in the previous report.

Reviewer #2 (Remarks to the Author):

The authors have thoroughly addressed my questions and those from the other referees.